# Token Coordinated Prompt Attention is Needed for Visual Prompting

Zichen Liu[1]  Xu Zou[2]  Gang Hua[3]  Jiahuan Zhou *[1]

## Abstract

Visual prompting techniques are widely used to efficiently fine-tune pretrained Vision Transformers (ViT) by learning a small set of shared prompts for all tokens. However, existing methods overlook the unique roles of different tokens in conveying discriminative information and interact with all tokens using the same prompts, thereby limiting the representational capacity of ViT. This often leads to indistinguishable and biased prompt-extracted features, hindering performance. To address this issue, we propose a plug-and-play Token Coordinated Prompt Attention (TCPA) module, which assigns specific coordinated prompts to different tokens for attention-based interactions. Firstly, recognizing the distinct functions of CLS and image tokens-global information aggregation and local feature extraction, we disentangle the prompts into CLS Prompts and Image Prompts, which interact exclusively with CLS tokens and image tokens through attention mechanisms. This enhances their respective discriminative abilities. Furthermore, as different image tokens correspond to distinct image patches and contain diverse information, we employ a matching function to automatically assign coordinated prompts to individual tokens. This enables more precise attention interactions, improving the diversity and representational capacity of the extracted features. Extensive experiments across various benchmarks demonstrate that TCPA significantly enhances the diversity and discriminative power of the extracted features. The code is available at https://github.com/zhoujiahuan1991/ICML2025-TCPA.

[1]Wangxuan Institute of Computer Technology, Peking University, Beijing, China [2]School of Artificial Intelligence and Automation, Huazhong University of Science and Technology, Wuhan, China [3]Amazon.com, Inc, Bellevue, WA, USA. Correspondence to: Jiahuan Zhou <jiahuanzhou@pku.edu.cn>.

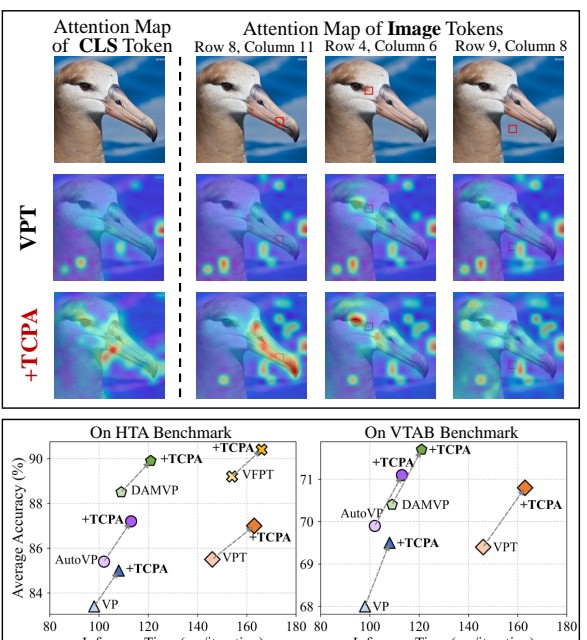

*Figure 1.* Above: Visualization of the attention map. The existing visual prompting method VPT (Jia et al., 2022) learns the same prompts for all tokens, resulting in extracted information that lacks distinguishability and comprehensiveness. Our TCPA selects corresponding prompts for different tokens and performs attention interaction, thereby enhancing the diversity and discriminability of the extracted information. Below: Comparison of time overhead and performance.

## 1. Introduction

In recent years, the pretraining-finetuning strategy has become a foundational paradigm in the deep learning field, significantly advancing the progress of various multi-media technologies (Jang et al., 2019; Guo et al., 2019; Iofinova et al., 2022; Xu et al., 2025; Li & Zhou, 2025; Yao et al., 2025). However, as the sizes of models and datasets have rapidly exploded, such a popular paradigm has faced critical challenges due to its high storage and computational costs (Jia et al., 2022). Addressing this, recent research (He et al., 2020; Cai et al., 2020; Zhang et al., 2020; Han et al., 2023) has focused on efficiently adapting pretrained models to specific downstream tasks. Among them, visual prompt-

ing has emerged as a leading player (Bahng et al., 2022; Jia et al., 2022; Huang et al., 2023) by introducing a minimal set of learnable prompts into the latest vision transformer (ViT) without retraining the original model parameters.

Existing visual prompting methods can be primarily categorized into two branches. Various works involve adding learnable prompts directly to the input sample itself, guiding the model to focus on discriminative information at the input-level (Bahng et al., 2022; Chen et al., 2023; Huang et al., 2023; Tsao et al., 2024). Besides, another branch introduces learnable tokens as prompts incorporated into each self-attention layer in ViT (Jia et al., 2022; Han et al., 2023; Yoo et al., 2023; Wang et al., 2024b). They aim to continuously prompt the model throughout the entire feature extraction process, facilitating the extraction of discriminative features. However, these methods usually learn and leverage the same prompt for all tokens without considering the different functionalities of CLS and image tokens, as well as the varying discriminative information conveyed by different image tokens. Consequently, this leads to different tokens focusing on similar regions and extracting biased discriminative information as shown in Figure 1, thereby limiting the representation ability of ViT.

To address the above issues, we introduce a plug-and-play Token Coordinated Prompt Attention (TCPA) module. It assigns specific coordinated prompts to different tokens for targeted attention-based interactions, allowing each prompt to contribute effectively to the extraction of comprehensive and discriminative information. Specifically, considering that CLS tokens and image tokens focus on global information aggregation and local feature extraction, respectively, we design CLS prompts and Image prompts for the CLS token and image tokens. These prompts interact exclusively with CLS tokens and image tokens within the attention blocks, thereby enhancing the discriminability of the extracted features. Furthermore, since different image tokens correspond to distinct image patches and the information they need to extract varies, we further disentangle CLS prompts and Image prompts into a CLS prompt Pool and an Image prompt Pool, each composed of multiple prompts. Token-coordinated prompts are automatically assigned to each token, improving the diversity of discriminative information in the extracted features.

To sum up, the main contributions of this work are: (1) To address the issues in existing visual prompting methods, we introduce a plug-and-play Token Coordinated Prompt Attention (TCPA) module, which assigns specific prompts to different tokens for targeted attention-based interactions, allowing each prompt to contribute effectively to the extraction of comprehensive and discriminative information. (2) Considering the differences in the information extracted by CLS and image tokens, as well as among different image to-

kens, we first disentangle the prompts into CLS prompts and Image prompts. We then match corresponding prompts to different tokens, fostering coordinated interactions between tokens and prompts and enhancing the discriminative ability of the extracted features. (3) Extensive experiments on various benchmarks show that TCPA consistently enhances the performance of existing state-of-the-art visual prompting methods.

## 2. Related Work

### 2.1. Parameter-Efficient Fine-Tuning

Vision Transformer (ViT) has made significant strides in computer vision research (Dosovitskiy et al., 2020; Liu et al., 2021b; Arnab et al., 2021; Chen et al., 2021a; Wang et al., 2021). However, the continuously increasing model sizes and datasets pose challenges in fully fine-tuning pretrained ViT models for downstream tasks, leading to substantial storage and computational cost. Consequently, recent studies (Zhang et al., 2020; Jia et al., 2022; Han et al., 2023) have shifted focus towards reducing the number of trainable parameters to streamline the fine-tuning process, broadly categorized as *partial tuning-based*, *extra module-based*, and *prompt learning-based* approaches.

Partial tuning-based methods (Yosinski et al., 2014; He et al., 2020; Noroozi & Favaro, 2016; Zhang et al., 2016) aim to retain most of the pretrained backbone while fine-tuning a smaller subset of parameters. Although straightforward and easy to implement, these methods often exhibit a noticeable performance gap compared to full fine-tuning (Chen et al., 2021b). On the other hand, extra module-based approaches (Rebuffi et al., 2017; Zhang et al., 2020; Cai et al., 2020; Pfeiffer et al., 2020; Zaken et al., 2021) introduce additional learnable plug-in architectures to fine-tune the pretrained model. However, these approaches are often tailored to specific architectures, limiting their applicability to other models. Moreover, the introduction of additional learnable parameters poses practical challenges, making them less feasible in real-world scenarios (Jia et al., 2022; Han et al., 2023).

### 2.2. Prompt Learning

Prompt learning techniques initially emerged in the realm of natural language processing (NLP), involving the incorporation of a small set of learnable soft prompts into input texts to customize language models for specific downstream tasks (Li & Liang, 2021; Liu et al., 2021a). Recent research has extended prompt learning to visual tasks, known as visual prompt tuning or visual prompting (Bahng et al., 2022; Jia et al., 2022; Han et al., 2023; Liu et al., 2024c;a;b). Compared to partial tuning-based and extra module-based methods, visual prompting-based approaches introduce sig-

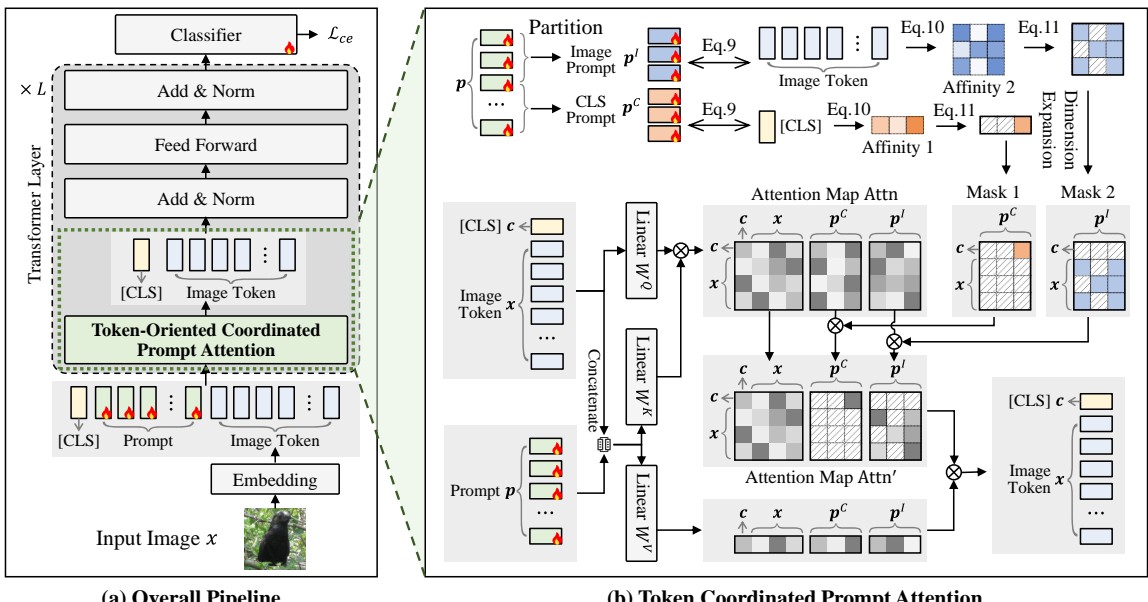

*Figure 2.* The overall pipeline of our proposed TCPA. For each input sample, embeddings for each image patch are first obtained through the embedding layer. Then, CLS and image tokens adaptively select appropriate prompts from the corresponding CLS and Image Prompt Pools and generate a binary mask. This binary mask is then fed into the attention module to mask certain values in the attention map, enabling attention-based interactions between different tokens and different prompts.

nificantly fewer additional parameters and exhibit superior compatibility with models of various architectures (Jia et al., 2022; Han et al., 2023).

Specifically, existing visual prompting methods can be mainly categorized into two types based on the location where prompts are applied: those added on the input image and those added within the token sequence. Prompting methods added on the input image overlay learnable visual prompts onto the original image to adjust pretrained models from the input level, enabling them to adapt to downstream tasks (Bahng et al., 2022; Huang et al., 2023; Chen et al., 2023; Wang et al., 2023; Tsao et al., 2024). These methods adjust pretrained models only in the input image level, making them adaptable to different network structures. However, since visual prompts are not used in the middle layers of the network, the representation capacity of prompts in such methods is limited, resulting in limited performance.

Another category of visual prompting involves introducing learnable tokens into the intermediate layers of the model, which undergos self-attention along with CLS and image tokens, thereby extracting discriminative features (Jia et al., 2022; Han et al., 2023; Yoo et al., 2023; Wang et al., 2024a). For instance, VPT (Jia et al., 2022) introduces learnable tokens at every layer of the vision transformer, allowing for individual adjustments across layers to better suit downstream tasks. These methods continuously provide prompts during the feature extraction process of the model. However,

they use the same prompts for all tokens without considering the distinct roles of CLS and image tokens, as well as the differences in discriminative information extracted by various image tokens. This results in the features extracted by different tokens being neither distinguishable nor comprehensive, which limits the model's performance.

## 3. Token Coordinated Prompt Attention

In this section, we illustrate the proposed *TCPA* in detail, and the overall pipeline is depicted in Figure 2.

### 3.1. Notations

In the architecture of a pretrained Vision Transformer (ViT) backbone, denoted as $\mathcal{M}$, there are $L$ instances of MSA (Multi-Head Self-Attention) blocks, symbolized as $\{\mathcal{B}_j\}_{j=1}^{L}$. Each block, $\mathcal{B}_j$, integrates multi-head self-attention with feed-forward networks, incorporating both LayerNorm and residual pathways for enhanced processing. When processing an input image $\boldsymbol{x}$, with dimensions $\mathbb{R}^{H \times W \times C}$, this image is partitioned into $N$ patches of uniform size $\{\boldsymbol{x}_i \in \mathbb{R}^{h \times w \times C}\}_{i=1}^{N}$. Here, $(H, W)$ represents the size of $\boldsymbol{x}$, $C$ is the channel count, and $(h, w)$ denotes the size of each patch $\boldsymbol{x}_i$. The transformation of each patch $\boldsymbol{x}_i$ into a $D$-dimensional feature space is given by:

$$\boldsymbol{h}_i^1 = \mathcal{E}(\boldsymbol{x}_i), \qquad (1)$$

where $\boldsymbol{h}_i^1 \in \mathbb{R}^D$, and $\mathcal{E}(\cdot)$ is the embedding layer of $\mathcal{M}$. These embedded patches, $\{\boldsymbol{h}_i^1\}_{i=1}^N$, along with a classification (CLS) token $\boldsymbol{c}_1 \in \mathbb{R}^D$, are sequentially processed through the $L$ MSA blocks $\{\mathcal{B}_j\}_{j=1}^L$. The operation can be summarized as follows:

$$[\boldsymbol{c}_{j+1}, \boldsymbol{h}_1^{j+1}, \cdots, \boldsymbol{h}_N^{j+1}] = \mathcal{B}_j([\boldsymbol{c}_j, \boldsymbol{h}_1^j, \cdots, \boldsymbol{h}_N^j]), \quad (2)$$

where the "[ ]" denotes the concatenation of the vectors. The final classification is conducted by passing the last MSA block's output, $\boldsymbol{c}_{L+1}$, through a classifier $\mathcal{H}$:

$$\boldsymbol{y} = \mathcal{H}(\boldsymbol{c}_{L+1}) \quad (3)$$

### 3.2. Coordinated Prompt Attention of CLS and Image Tokens

In the vision transformer, an image $\boldsymbol{x}$ is initially divided into numerous small patches, which are then converted into corresponding image tokens via an embedding layer. During the attention process, image tokens continuously extract discriminative information from the input sample $\boldsymbol{x}$. This information is subsequently aggregated through a CLS token, summarizing the insights gathered from all image tokens for final classification. It is evident that the role of image tokens is to extract discriminative information, whereas the CLS token's purpose is to aggregate this information and facilitate classification, highlighting the distinct functions of these two types of tokens. Hence, we design Coordinated Prompt Attention of CLS and Image Tokens, which disentangles prompts for CLS and image tokens, aiding them in better fulfilling their respective functions.

Specifically, we disentangle prompts for CLS and image tokens, denoted as $\mathbf{P}^c$ and $\mathbf{P}^i$, respectively:

$$\mathbf{P}^c = \{\boldsymbol{p}_j^c\}_{j=1}^L, \quad (4)$$

$$\mathbf{P}^i = \{\boldsymbol{p}_j^i\}_{j=1}^L, \quad (5)$$

where $\boldsymbol{p}_j^c \in \mathbb{R}^{L_p \times D}$ and $\boldsymbol{p}_j^i \in \mathbb{R}^{L_p \times D}$ are CLS and image prompt for $j$-th MSA block $\mathcal{B}_j$.

Then we feed the CLS token $\boldsymbol{c}_j$, image tokens $(\boldsymbol{h}_1^j, \cdots, \boldsymbol{h}_N^j)$, and the CLS prompt $\boldsymbol{p}_j^c$ together into the MSA block $\mathcal{B}_j$, obtaining the corresponding output:

$$\begin{aligned}[\boldsymbol{c}_{j+1}, \boldsymbol{p}_{j+1}^{c,d}, \boldsymbol{h}_1^{j+1,d}, \cdots, \boldsymbol{h}_N^{j+1,d}] \\ = \mathcal{B}_j([\boldsymbol{c}_j, \boldsymbol{p}_j^c, \boldsymbol{h}_1^j, \cdots, \boldsymbol{h}_N^j]), \end{aligned} \quad (6)$$

where the subscript $d$ in $\boldsymbol{p}_{j+1}^{c,d}, \boldsymbol{h}_1^{j+1,d}, \cdots, \boldsymbol{h}_N^{j+1,d}$ indicates that these outputs will be discarded and not utilized by subsequent MSA layers. In the equation above, only the output CLS token continues to be used, and therefore, the CLS prompt only affects the CLS token, not the image tokens.

Similarly, we input the image prompt $\boldsymbol{p}_j^i$ along with the CLS token $\boldsymbol{c}_j$ and image tokens $(\boldsymbol{h}_1^j, \cdots, \boldsymbol{h}_N^j)$ into the MSA block $\mathcal{B}_j$, obtaining the output for the image tokens:

$$[\boldsymbol{c}_{j+1}^d, \boldsymbol{p}_{j+1}^{i,d}, \boldsymbol{h}_1^{j+1}, \cdots, \boldsymbol{h}_N^{j+1}] = \mathcal{B}_j([\boldsymbol{c}_j, \boldsymbol{p}_j^i, \boldsymbol{h}_1^j, \cdots, \boldsymbol{h}_N^j]), \quad (7)$$

where the subscript $d$ in $\boldsymbol{c}_{j+1}^d, \boldsymbol{p}_{j+1}^{i,d}$ indicates that these outputs will be discarded. Through the equation above, we obtain the output for the image tokens $(\boldsymbol{h}_1^{j+1}, \cdots, \boldsymbol{h}_N^{j+1})$, which, together with the previously obtained output of the CLS token $\boldsymbol{c}_{j+1}$, serves as the input for the next MSA block $\mathcal{B}_{j+1}$.

### 3.3. Coordinated Prompt Attention of Different Image Tokens

In the previous text, we disentangle prompts into CLS and image prompts based on their distinct roles. However, in vision transformers, different image tokens correspond to different image patches with varying discriminative information. Using the same prompts for all image tokens can make the extracted features indistinguishable and biased. Thus, we further introduce coordinated prompt attention of different image tokens to enhance the pretrained model's ability to extract rich, discriminative information from the input image.

To simplify notation, in the following discussion, we will not explicitly differentiate prompts from different layers. It's important to note that while the parameters of prompts vary across layers, the processing method for prompts remains consistent throughout. Specifically, we disentangle the image prompt $\boldsymbol{p}^i$ into a image prompt pool $\mathcal{P}^i$ composed of multiple image prompts:

$$\mathcal{P}^i = \{(\boldsymbol{p}_k^i, \boldsymbol{\kappa}_k^i)\}_{k=1}^{N_i}, \quad (8)$$

where $\boldsymbol{\kappa}_k^i$ represents the learnable indicator corresponding to the prompt $\boldsymbol{p}_k^i$, used for selecting the prompt based on the image token.

For each image token $\boldsymbol{h}_m^j$, the distance between $\boldsymbol{h}_m^j$ and the prompt indicator $\boldsymbol{\kappa}_k^i$ can be measured via a cosine distance $\mathcal{S}(\cdot, \cdot)$:

$$\begin{aligned}\mathcal{S}(\boldsymbol{h}_m^j, \boldsymbol{\kappa}_k^i) &= 1 - \cos(\boldsymbol{h}_m^j, \boldsymbol{\kappa}_k^i) \\ &= 1 - \frac{\boldsymbol{h}_m^j \cdot \boldsymbol{\kappa}_k^i}{\left\|\boldsymbol{h}_m^j\right\|_2 \cdot \left\|\boldsymbol{\kappa}_k^i\right\|_2}.\end{aligned} \quad (9)$$

Through the above process, we obtain the affinity matrix $\mathbf{A} \in \mathbb{R}^{N \times N_i}$ between image tokens and different image prompts, where $\mathbf{A}_{m,k} = \mathcal{S}(\boldsymbol{h}_m^j, \boldsymbol{\kappa}_k^i)$. Then, we perform binarization on the matrix $\mathbf{A}$, setting the top $K_i$ largest elements in each row to 1 while assigning 0 to all other elements. Specifically, the binarized matrix $\hat{\mathbf{A}} \in \{0, 1\}^{N \times N_i}$

is defined as follows:

$$\hat{\mathbf{A}}_{m,k} = \mathbb{I}\left(\sum_{s=1}^{K_i} \mathbb{I}(k = \pi_m(s)) > 0\right), \qquad (10)$$

where $\mathbb{I}(\cdot)$ denotes the indicator function, which takes the value of 1 if the condition inside holds and 0 otherwise; $\pi_m$ represents the index sequence obtained by sorting the elements of the $i$-th row of matrix $\mathbf{A}$ in descending order, i.e., satisfying $\mathbf{A}_{m,\pi_m(1)} \geq \mathbf{A}_{m,\pi_m(2)} \geq \cdots \geq \mathbf{A}_{m,\pi_m(n)}$; and $\pi_m(s)$ corresponds to the column index of the $s$-th largest element after sorting. Through this operation, we obtain a binary mask matrix $\mathbf{M}$, which selects the top $K_i$ largest elements in each row of $\mathbf{A}$ while suppressing the influence of other irrelevant elements. Then, we align the dimensions of $\hat{\mathbf{A}}$ with the dimensions of the attention map to obtain the final image token mask:

$$\mathbf{M}^i_{m,k} = \begin{cases} 0, & \text{if } m = 0 \\ \hat{\mathbf{A}}_{m+1,k}, & \text{otherwise} \end{cases} \qquad (11)$$

To guide the CLS token corresponding to different samples in better aggregating global information, we also disentangle the CLS prompt $\boldsymbol{p}^c$ into a CLS prompt pool $\mathcal{P}^c = \{(\boldsymbol{p}^c_k, \boldsymbol{\kappa}^c_k)\}_{k=1}^{N_c}$. In a similar manner to the image tokens, we can obtain the affinity matrix between the CLS token and the CLS prompts. Then, by further binarizing and expanding the dimensions, we obtain the mask $\mathbf{M}^c$ corresponding to the CLS token.

In Vision Transformers, the core operation of the attention module is:

$$\text{Attn} = \text{Softmax}\left(\frac{QK^T}{\sqrt{d_k}}\right), \qquad (12)$$

where $\text{Attn}$ is the attention map. We concatenate the two masks, $\mathbf{M}^c$ and $\mathbf{M}^i$, corresponding to the CLS token and the image token, then expand them to the same dimensions as the attention map $\text{Attn}$, resulting in the final mask $\mathbf{M}$. Finally, we perform an element-wise multiplication between $\text{Attn}$ and the mask $\mathbf{M}$ to obtain the updated attention map for subsequent operations:

$$\text{Attn}' = \text{Attn} \odot \mathbf{M}, \qquad (13)$$

where $\odot$ denotes the element-wise multiplication.

Although our TCPA selects specific prompts for each token, the additional computational overhead is limited only to the calculation of cosine distance for prompt selection and the self-attention process, with no increase in the computation of the feed-forward network. Furthermore, we can achieve the effect of multiple groups of tokens undergoing attention separately through a single attention computation by utilizing token masking. In TCPA, we compute the attention

weights of all tokens and prompts, $\text{Attn} = \text{Softmax}\left(\frac{QK^T}{\sqrt{d_k}}\right)$. Then, based on the token-prompt matching, we generate a binary mask matrix $\mathbf{M}$ and compute $\text{Attn} \cdot \mathbf{M} \cdot V$. This approach calculates the attention weights only once, enabling efficient interaction between different tokens and prompts.

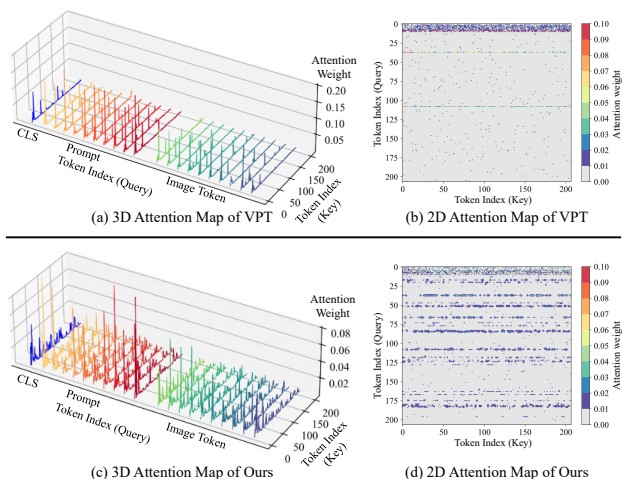

(a) 3D Attention Map of VPT

(b) 2D Attention Map of VPT

(c) 3D Attention Map of Ours

(d) 2D Attention Map of Ours

*Figure 3.* 3D and 2D attention map of existing visual prompting method VPT (Jia et al., 2022) and Ours.

### 3.4. Overall Optimization

As mentioned above, our TCPA introduces only a few additional parameters: CLS prompt pool $\mathcal{P}^c$ and image prompt pool $\mathcal{P}^i$. Following (Jia et al., 2022), during training, we maintain the pretrained model's encoder frozen while allowing only the classification head to be trainable. We denote all learnable parameters as $\boldsymbol{\Phi} = \{\mathcal{P}^c, \mathcal{P}^i, \mathcal{H}\}$. The optimization objective is as follows:

$$\arg\min_{\boldsymbol{\Phi}} \mathcal{L}_{ce}(\boldsymbol{y}, y_{gt}) + \lambda_i \sum \mathcal{S}(\boldsymbol{h}_m, \boldsymbol{\kappa}^i_m) + \lambda_c \sum \mathcal{S}(\boldsymbol{c}_j, \boldsymbol{\kappa}^c_j),$$
$$(14)$$

where $\mathcal{L}_{ce}$ is cross-entropy loss, $y_{gt}$ is the label of image $\boldsymbol{x}$, $\lambda_i$ and $\lambda_c$ are weighting parameters.

### 4. Discussion and Analysis

To further analyze and validate the effectiveness of our method, this section provides mathematical and experimental analysis of why existing visual prompting methods extract discriminative information that is singular and insufficient, while our TCPA method extracts comprehensive discriminative information.

**Theorem 4.1.** *Self-attention is low rank. (Proved in (Wang et al., 2020)). Let $A \in \mathbb{R}^{n \times n}$ be a self-attention matrix, and $v \in \mathbb{R}^n$ be a column vector of value matrix $V$. Then, there*

*Table 1.* The comparison results on HTA benchmark. *Partial*, *Extra*, and *Prompting* represent partial tuning-based, extra module-based, and prompt learning-based methods respectively.

| | Methods | | DTD | CUB | Bird | Dog | Flower | Food | Cifar100 | Cifar10 | GTSRB | SVHN | Avg |
|---|---|---|---|---|---|---|---|---|---|---|---|---|---|
| | Full | - | 64.3 | 87.3 | 82.7 | 89.4 | 98.8 | 84.9 | 68.9 | 97.4 | 97.1 | 87.4 | 85.8 |
| *Partial* | Linear | - | 63.2 | 85.3 | 75.9 | 86.2 | 97.9 | 84.4 | 63.4 | 96.3 | 68.0 | 36.6 | 75.7 |
| | Partial | *NeurIPS'14* | 70.1 | 85.6 | 77.8 | 85.5 | 98.2 | 83.8 | 78.0 | 95.0 | 89.3 | 82.4 | 84.6 |
| | MLP | *CVPR'20* | 66.2 | 85.1 | 77.3 | 84.9 | 97.9 | 84.6 | 77.5 | 93.2 | 71.8 | 60.5 | 79.9 |
| *Extra* | Bias | *NeurIPS'17* | 69.8 | 88.4 | 84.2 | 91.2 | 98.8 | 86.2 | 82.9 | 96.9 | 89.9 | 82.5 | 87.1 |
| | Sidetune | *ECCV'20* | 57.7 | 84.7 | 75.8 | 85.8 | 96.9 | 78.7 | 68.8 | 90.4 | 90.9 | 80.5 | 81.0 |
| | Adapter | *NeurIPS'20* | 62.7 | 87.1 | 84.3 | 89.8 | 98.5 | 86.0 | 74.2 | 97.7 | 91.1 | 36.3 | 80.8 |
| | Former | *NeurIPS'22* | 64.2 | 87.3 | 84.1 | 88.1 | 98.4 | 85.7 | 79.4 | 96.5 | 91.7 | 83.0 | 85.8 |
| *Prompting* | $E^2$VPT | *ICCV'23* | 66.8 | 88.4 | 84.2 | 91.3 | 99.0 | 84.0 | 80.4 | 97.1 | 91.0 | 79.2 | 86.1 |
| | LION | *AAAI'24* | - | - | - | 83.6 | 90.5 | - | 65.4 | 90.8 | - | - | - |
| | VP | *arXiv'22* | 59.5 | 84.6 | 77.7 | 84.5 | 97.7 | 80.5 | 78.7 | 94.2 | 89.4 | 87.6 | 83.4 |
| | **+TCPA** | *This Paper* | 62.3(+2.8) | 86.7(+2.1) | 78.7(+1.0) | 88.6(+3.1) | 99.3(+1.6) | 81.9(+1.4) | 81.4(+2.7) | 95.1(+0.9) | 90.7(+1.3) | 89.6(+2.0) | 85.4(+2.0) |
| | VPT | *ECCV'22* | 65.8 | 88.5 | 84.2 | 90.2 | 99.0 | 83.3 | 78.8 | 96.8 | 90.7 | 78.1 | 85.5 |
| | **+TCPA** | *This Paper* | 67.6(+1.8) | 90.7(+2.2) | 85.5(+1.3) | 91.7(+1.5) | 99.2(+0.2) | 84.8(+1.5) | 79.9(+1.1) | 98.9(+2.1) | 92.1(+1.4) | 79.4(+1.3) | 87.0(+1.5) |
| | DAMVP | *CVPR'23* | 73.1 | 87.5 | 82.1 | 92.3 | 99.2 | 86.9 | 88.1 | 97.3 | 90.6 | 87.9 | 88.5 |
| | **+TCPA** | *This Paper* | 74.3(+1.2) | 88.9(+1.4) | 82.7(+0.6) | 93.6(+1.3) | 99.3(+0.1) | 88.7(+1.8) | 89.7(+1.6) | 97.8(+0.5) | 93.8(+3.2) | 90.4(+2.5) | 89.9(+1.4) |
| | AutoVP | *ICLR'24* | 62.5 | 85.4 | 83.5 | 90.3 | 90.4 | 82.3 | 77.9 | 95.2 | 93.1 | 92.9 | 85.4 |
| | **+TCPA** | *This Paper* | 65.0(+2.5) | 88.0(+2.6) | 85.6(+2.1) | 92.5(+2.2) | 91.2(+0.8) | 83.6(+1.3) | 79.3(+1.4) | 97.4(+2.2) | 96.2(+3.1) | 93.5(+0.6) | 87.2(+1.8) |
| | VFPT | *NeurIPS'24* | 69.9 | 88.1 | 82.8 | 89.5 | 98.4 | 88.7 | 90.4 | 97.4 | 92.1 | 94.4 | 89.2 |
| | **+TCPA** | *This Paper* | 71.2(+1.3) | 89.5(+1.4) | 83.6(+0.8) | 91.5(+2.0) | 99.2(+0.8) | 89.2(+0.5) | 91.6(+1.2) | 98.4(+1.4) | 94.2(+2.1) | 95.6(+1.2) | 90.4(+1.2) |

*Table 2.* The comparison results on VTAB benchmark. Utilize the ViT-B/16 pretrained with supervised training on ImageNet-21k as the backbone.

| Methods | Natural | Specialized | Structured |
|---|---|---|---|
| Full | 75.9 | 83.4 | 47.6 |
| VP | 77.3 | 80.1 | 53.8 |
| **+TCPA** | 78.1(+0.8) | 82.6(+2.5) | 55.4(+1.6) |
| VPT | 78.5 | 82.4 | 55.0 |
| **+TCPA** | 79.7(+1.2) | 84.3(+1.9) | 56.2(+1.2) |
| DAMVP | 79.1 | 83.4 | 56.2 |
| **+TCPA** | 80.4(+1.3) | 85.5(+2.1) | 57.1(+0.9) |
| AutoVP | 78.4 | 83.1 | 55.8 |
| **+TCPA** | 79.3(+0.9) | 85.2(+2.1) | 56.9(+1.1) |

*exists a low-rank matrix $\hat{A} \in \mathbb{R}^{n \times n}$ satisfying*

$$Pr(\|\hat{A}v^T - Av^T\| < \epsilon\|Av^T\|) > 1 - o(1), \qquad (15)$$

*where the rank of $\hat{A}$ is bounded, i.e., $rank(A) = \Theta(log(n))$.*

**Theorem 4.2.** *Self-attention is low-rank after prompting. (Proved in (Kim et al., 2024)). For any low-rank matrices $\hat{A}_n \in \mathbb{R}^{n \times n}$ and $\hat{A}_{n+m} \in \mathbb{R}^{(n+m) \times (n+m)}$ satisfying $Pr(\|\hat{A}v^T - Av^T\| < \epsilon\|Av^T\|) > 1 - o(1)$, we have*

$$rank(\hat{A}_{n+m} - rank(\hat{A}_n) = O(log(m)), \qquad (16)$$

*where $m$ is the number of prompts.*

Through the above two theorems, we can see that the self-attention matrix in existing prompt learning methods is low-rank. This indicates that different prompts in these methods tend to focus on the same image regions. To further demonstrate this, we visualize the attention maps of the existing visual prompting method VPT in both 3D and 2D. As shown in Figure. 3, the attention regions of prompts in conventional visual prompting methods are highly similar, leading to CLS and image tokens extracting nearly identical features.

In contrast, our proposed TCPA module enhances more diverse attention across prompts, CLS tokens, and image tokens. This is because our method selects different prompts for different tokens and performs attention-based interactions, thereby encouraging the model to extract more diverse and comprehensive discriminative information.

## 5. Experiments

### 5.1. Datasets

Building upon (Jia et al., 2022; Huang et al., 2023), the experiments are conducted on HTA (Huang et al., 2023) benchmark, including: DTD (Cimpoi et al., 2014), CUB-200-2011 (Wah et al., 2011), NABirds (Horn et al., 2015), Stanford Dogs (Khosla et al., 2011), Oxford Flowers (Nilsback & Zisserman, 2008), Food101 (Bossard et al., 2014), Cifar100 (Krizhevsky et al., 2009), Cifar10 (Krizhevsky et al., 2009), GTSRB (Stallkamp et al., 2012), and SVHN (Netzer et al., 2011).

Moreover, following (Jia et al., 2022; Han et al., 2023), more experiments are conducted on the VTAB benchmark (Zhai et al., 2019) which includes 19 visual tasks. These tasks are categorized into three groups: *Natural*, for routine image recognition; *Specialized*, for domain-specific applications such as medical imaging; and *Structured*, for the analysis of intricate scenes, like 3D object recognition.

## 5.2. Comparison Methods

We compare TCPA with the parameter-efficient fine-tuning and visual prompting methods. We also report the fully-tuning results as a baseline. Specifically, the parameter-efficient fine-tuning methods include the partial tuning-based models (Linear (Iofinova et al., 2022), Partial (Yosinski et al., 2014), MLP (He et al., 2020)) and the extra module-based ones (Sidetune (Rebuffi et al., 2017), Bias (Zhang et al., 2020), Adapter (Cai et al., 2020), Adapt-Former (Chen et al., 2022)). For visual prompting methods, various latest visual prompting methods such as VP (Bahng et al., 2022), VPT (Jia et al., 2022), DAMVP (Huang et al., 2023), Yoo et al (Yoo et al., 2023), E$^2$VPT (Han et al., 2023), LION (Wang et al., 2023), AutoVP (Tsao et al., 2024) and VFPT (Zeng et al., 2024) are evaluated.

## 5.3. Implementation Details

To fully validate the effectiveness of our proposed TCPA, we implement TCPA based on several representative visual prompting methods from recent years. For the token-level methods VPT (Jia et al., 2022) and VFPT (Zeng et al., 2024), we replace their learnable prompt tokens with our TCPA. For the input-level prompting methods VP (Bahng et al., 2022), DAMVP (Huang et al., 2023), and AutoVP (Tsao et al., 2024), while retaining their prompts added to the input images, we introduce our TCPA at the token level. The ViT-B/16 (Dosovitskiy et al., 2020) supervised by ImageNet-21k (Deng et al., 2009) is used as the backbone. Following DAMVP (Huang et al., 2023), we train for 100 epochs on all datasets. The AdamW (Loshchilov & Hutter, 2017) optimizer and cosine annealing are used for optimization. The weighting parameters $\lambda_i$ and $\lambda_c$ are set to 0.5. The length of the size of CLS prompt pool $N_c$ and size of image prompt pool $N_i$ are set to 10 and 20 respectively.

## 5.4. Comparison with State-of-the-arts

We first conduct experiments on HTA using the ImageNet-21k supervised ViT-B/16 (Dosovitskiy et al., 2020) as the pretrained model. As shown in Table 1, after integrating TCPA, in comparison to DAMVP, DAMVP+TCPA shows average improvements of 1.4% across all ten datasets. Similar enhancements are also observed when applied to other methods: VP+TCPA shows an average increase of 0.9%-2.8% across the ten datasets, VPT+TCPA improved by 0.2%-2.2%, AutoVP+TCPA improved by 0.6%-3.1%, and VFPT+TCPA also improved by 0.5%-2.0%. This can be primarily attributed to TCPA's explicit disentanglement of prompts based on the distinct roles of CLS and image tokens and their difference in the attention mechanism, allowing for more thorough learning of downstream task knowledge and facilitating comprehensive extraction of discriminative information, thereby boosting model performance.

*Table 3.* The influence of components in TCPA. "✓" represent with this component. R-TCPA represents the coordinated prompt attention of CLS and image tokens. T-TCPA represents the coordinated prompt attention of different image tokens.

| Components | | Datasets | | |
|---|---|---|---|---|
| R-TCPA | T-TCPA | CUB | Dog | GTSRB |
| - | - | 88.1 | 89.5 | 92.1 |
| ✓ | - | 88.9 | 90.2 | 93.0 |
| ✓ | ✓ | 89.5 | 91.5 | 94.1 |

To further validate the effectiveness of our TCPA, following (Jia et al., 2022; Han et al., 2023), we also conduct experiments on VTAB (Zhai et al., 2019) benchmark. As shown in Table 2, compared to VPT, the integration of TCPA results in performance improvements of 1.2%, 1.9%, and 1.2% across downstream tasks in groups Natural, Specialized, and Structured, respectively. Moreover, TCPA consistently enhances performance based on VP, DAMVP, and AutoVP. Specifically, VP+TCPA shows improvements of 0.8%, 2.5%, and 1.6%, DAMVP+TCPA shows improvements of 1.3%, 2.1%, and 0.9%, and AutoVP+TCPA also exhibits performance enhancements of 0.9%, 2.1%, and 1.1% across the same groups of downstream tasks. This further demonstrates TCPA's robustness across a variety of downstream tasks.

## 5.5. Ablation

### 5.5.1. INFLUENCE OF DIFFERENT COMPONENTS

To further validate the effectiveness of each component we proposed, we conduct ablation studies on the two main components of TCPA: R-TCPA and T-TCPA. When none of the modules are utilized, the method degenerates to the original VPT approach. Conversely, employing all three modules constitutes the complete VPT+TCPA method. As shown in Table 3, the introduction of the R-TCPA module leads to a performance increase of 0.8%-0.9%. This improvement is attributed to R-TCPA's disentanglement of CLS and image prompts based on their distinct roles, guiding CLS and image tokens to fulfill different functions. Further incorporating T-TCPA allows different image tokens to adaptively select suitable prompts from the prompt pool, thoroughly capturing diverse discriminative information from the input sample, hence boosting model performance by an additional 0.6%-1.1%.

### 5.5.2. COMPUTATIONAL COST

To validate the efficiency of our proposed method, we conduct a computational cost analysis to compare our proposed method, TCPA, with existing visual prompting approaches. As shown in Table 4, TCPA introduces only a minimal additional time cost while enhancing model performance.

*Table 4.* Comparison of training time on CUB (seconds/epoch) with state-of-the-art.

| Methods | Training Time |
|---------|---------------|
| VP (Bahng et al., 2022) | 1.94 |
|   +TCPA | 2.24 |
| VPT (Jia et al., 2022) | 2.36 |
|   +TCPA | 2.41 |
| DAMVP (Huang et al., 2023) | 2.14 |
|   +TCPA | 2.27 |
| VFPT (Zeng et al., 2024) | 2.58 |
|   +TCPA | 2.65 |

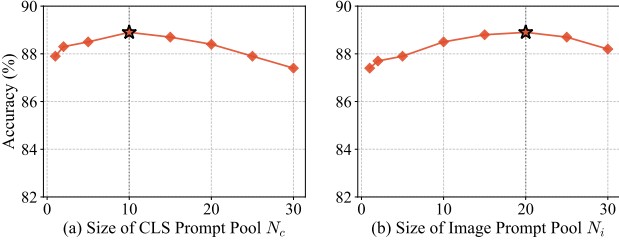

*Figure 4.* Influence of hyper-parameters (size of CLS prompt pool $N_c$, size of image prompt pool $N_i$) of TCPA on CUB.

Despite disentangling the prompts used for the CLS and image tokens, as well as between different image tokens, the increase in computational demand is negligible. The additional computational load from the prompt matching process in TCPA is primarily achieved through vector multiplication, making the increase in computational demand minimal. Although different prompts are used for different tokens, in the implementation, we input all prompts simultaneously into the attention mechanism for query-key computations. During the final computation with the attention mechanism's values, different masks are applied to different tokens to enable targeted attention interactions between specific tokens and prompts. This implementation strategy significantly reduces the computational overhead of our method.

### 5.5.3. INFLUENCE OF HYPER-PARAMETERS

As illustrated in Figure 4, we conducted ablation experiments on two hyperparameters introduced in TCPA: the size of the CLS prompt pool $N_c$ and the image prompt pool $N_i$. When the size of the prompt pool is small, there is lower diversity among prompts, leading to a higher overlap of prompts selected by different tokens. This results in the features extracted from different tokens becoming indistinguishable. Conversely, an excessive number of prompts in the pool increases the number of learnable parameters. Given the limited data in downstream tasks, this scenario

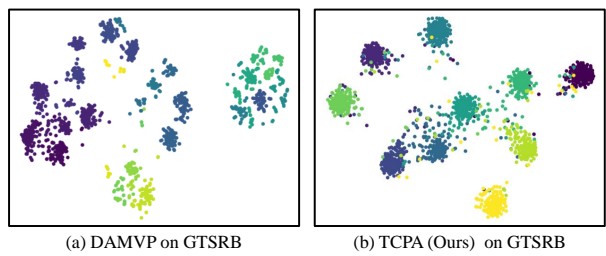

(a) DAMVP on GTSRB      (b) TCPA (Ours) on GTSRB

*Figure 5.* Feature t-SNE (Van der Maaten & Hinton, 2008) visualization results for our proposed TCPA and comparison method DAMVP (Huang et al., 2023) on GTSRB.

can lead to overfitting, which also degrades model performance. Optimal performance is achieved when the size of the prompt pool is moderate. Notably, the image prompt pool requires a larger size than the CLS prompt pool because image tokens exhibit greater variability compared to the CLS tokens used directly for classification, necessitating a broader range of prompts.

### 5.5.4. THE T-SNE VISUALIZATION OF EXTRACTED FEATURES

To further validate the effectiveness of our method, Figure 5 presents a t-SNE (Van der Maaten & Hinton, 2008) visualization of features obtained by TCPA and DAMVP (Huang et al., 2023). As shown in Figure 5, we visualize the features obtained by our proposed TCPA and DAMVP via t-SNE (Van der Maaten & Hinton, 2008). From the visualization results, it can be observed that the features extracted by DAMVP from samples of the same category are relatively scattered, and some are mixed with features from other categories. In contrast, features extracted by our TCPA from the same category are tightly clustered and display clear distinctiveness from features of other categories. This is attributed to our proposed token coordinated prompt attention, which can recognize more diverse and comprehensive discriminative characteristics of input images.

## 6. Conclusion

In this paper, we propose a novel plug-and-play Token Coordinated Prompt Attention (TCPA) module to enhance visual prompting for Vision Transformers. Unlike existing methods that learn the same prompts for all tokens, TCPA disentangles and adaptively assigns prompts to different CLS and image tokens based on their distinct roles, thereby improving feature diversity and discriminability. Specifically, we introduce CLS Prompts and Image Prompts to interact exclusively with CLS and image tokens, respectively, strengthening their individual representational capacities. Furthermore, TCPA leverages a matching function to dy-

namically allocate coordinated prompts to image tokens, enabling more precise and targeted attention interactions. By incorporating these mechanisms, TCPA effectively mitigates the limitations of conventional visual prompting, leading to richer, more diverse feature extraction and improved model performance, as demonstrated by experimental and visualization results.

# Acknowledgments

This work was supported by the National Key R&D Program of China (2024YFA1410000) and the National Natural Science Foundation of China (62376011).

# Impact Statement

This paper presents work whose goal is to advance the field of Machine Learning. There are many potential societal consequences of our work, none which we feel must be specifically highlighted here.

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

## A. More t-SNE Visualization Results of Extracted Features

To further validate the effectiveness of our method, we also visualized the features extracted on several other datasets using t-SNE (Van der Maaten & Hinton, 2008). As shown in Figure 6, compared to the existing method DAMVP, the visualization results of our proposed TCPA show tighter clustering of samples within the same category and better separability between different categories. This is due to the hierarchically disentangled visual prompting in TCPA, which disentangles prompts based on different roles and functions. Each prompt is tailored to effectively and comprehensively extract semantic information from the image samples, enhancing the discriminative ability of the extracted feature.

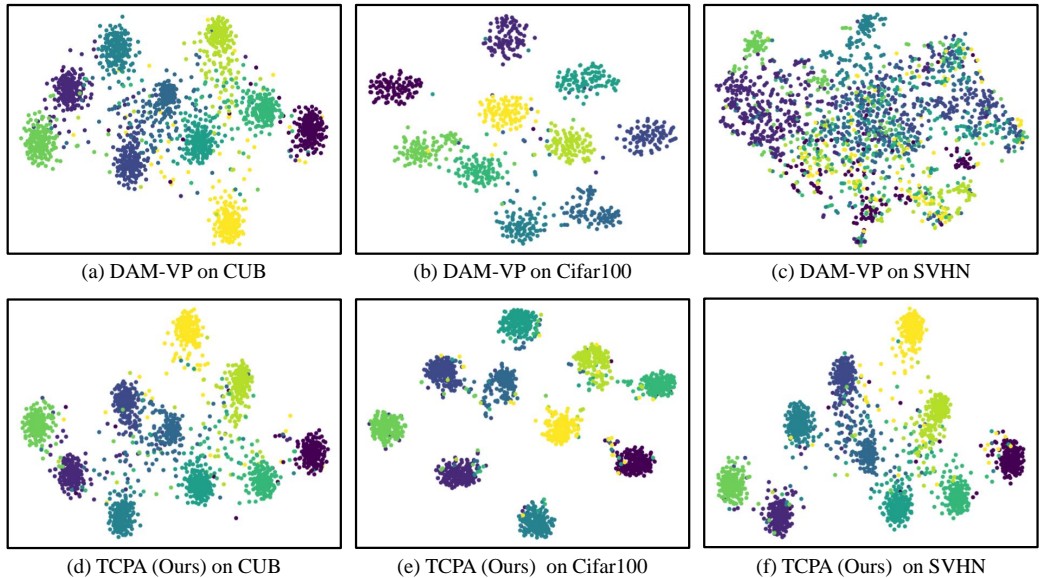

(a) DAM-VP on CUB        (b) DAM-VP on Cifar100        (c) DAM-VP on SVHN

(d) TCPA (Ours) on CUB        (e) TCPA (Ours) on Cifar100        (f) TCPA (Ours) on SVHN

*Figure 6.* Feature t-SNE (Van der Maaten & Hinton, 2008) visualization results for our proposed TCPA and comparison method DAMVP on CUB, Cifar100 and SVHN.

## B. More Attention Visualization Results

In Figure 7, we provide attention map visualizations for additional samples. As shown, existing methods, which learn the same prompt for all tokens, result in different tokens are indistinguishable and biased. In contrast, our proposed TCPA disentangles the prompts used for different tokens, thereby enhancing the diversity and discriminative ability of the features extracted by each token.

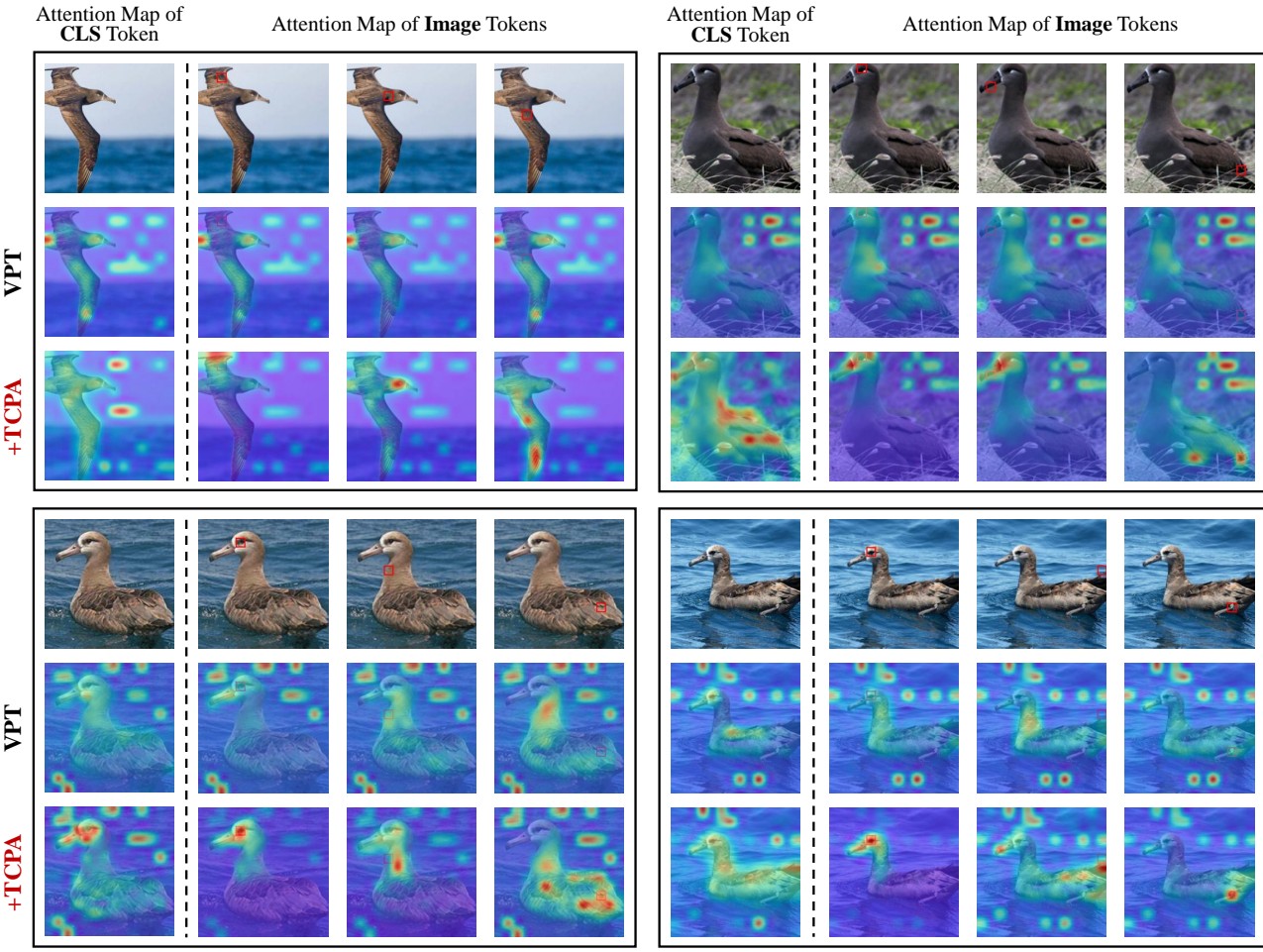

*Figure 7.* The attention map visualization results of CLS and image tokens from the existing visual prompting method VPT (Jia et al., 2022) and our TCPA are presented.

