# OpenReview forum: "Token Coordinated Prompt Attention is Needed for Visual Prompting"
_ICML.cc/2025/Conference — ICML 2025 poster_

### Official Review · Reviewer_2ZNj · 2025-03-03

**Overall Recommendation:** 4

**Summary:**

This paper proposes a Token Coordinated Prompt Attention (TCPA) module to enhance the effectiveness of visual prompting in Vision Transformers (ViT). Existing methods use shared prompts for all tokens, overlooking the distinct roles of CLS and image tokens, leading to limited representational capacity. TCPA addresses this by assigning CLS and image-specific prompts for targeted attention interactions, improving their discriminative abilities. A matching function further assigns coordinated prompts to individual image tokens to enhance feature diversity and representation. Experiments show that TCPA significantly improves feature diversity and performance.

## update after rebuttal
I have reviewed the authors' rebuttal as well as the comments from my fellow reviewers. I remain inclined to maintain my positive assessment and will keep my current rating.

**Claims And Evidence:**

The claims made in the submission are well-supported by clear and convincing evidence. The paper thoroughly validates the effectiveness of the proposed Token Coordinated Prompt Attention (TCPA) module through extensive experiments across multiple benchmarks. The authors provide detailed comparisons with state-of-the-art methods, demonstrating significant improvements in both feature diversity and overall performance. Additionally, ablation studies are conducted to isolate the contributions of CLS-specific and image-specific prompts, highlighting the effectiveness of each component.

**Essential References Not Discussed:**

No essential related works crucial for understanding the contributions of the paper have been omitted. The paper provides a comprehensive overview of related work in visual prompting, covering the core literature relevant to the proposed method. The authors have appropriately applied the relevant theories and provided a thorough introduction to them. These citations and discussions offer strong background support for understanding the key contributions of the paper.

**Experimental Designs Or Analyses:**

The experimental design and analysis in the paper are both reasonable and effective. The authors have selected appropriate comparative experiments to validate the proposed method, and the benchmarks and evaluation metrics used in the experiments are well-targeted, allowing for a comprehensive assessment of the model's performance. Additionally, ablation studies are included to verify the performance of each module. The paper also presents several visualization experiments, further supporting the theoretical claims made. These design choices significantly contribute to the reliability and validity of the research.

**Methods And Evaluation Criteria:**

The proposed methods and evaluation criteria (e.g., benchmark datasets) are appropriate and well-suited to the problem at hand. The chosen benchmark datasets are representative, providing valuable insights into the model's performance across various scenarios.

**Other Comments Or Suggestions:**

It is recommended that the authors include pseudocode of the method to help readers better understand the process and flow of the proposed approach. This would make it easier for others to reproduce or build upon the work.

**Other Strengths And Weaknesses:**

Strengths:
1. The proposed method is novel. The paper introduces Token Coordinated Prompt Attention for the first time, which changes the way tokens and prompts interact in prompt learning methods. By selecting specific prompts for each token to interact with, this approach helps the model extract rich and comprehensive discriminative information.
2. The paper is well-structured and coherent. The logical flow is smooth, and the writing is clear. The accompanying figures effectively illustrate and validate the points made in the paper, making the arguments more accessible and convincing.
3. The paper provides comprehensive comparative experiments. The authors conduct experiments across multiple benchmark datasets and integrate the proposed module into several existing prompt learning methods, achieving consistent performance improvements. This demonstrates the effectiveness and generalizability of the proposed approach across different scenarios.
4. The ablation studies are well-designed. The authors include a wide range of ablation and visualization experiments, which help readers understand the role of each module and provide a clear visualization of how the proposed TCPA influences the attention maps and feature extraction process of the model.

Weaknesses:
1. The authors only show examples of 2D and 3D attention maps for a single sample. Providing additional examples across a wider range of samples would further strengthen the argument and offer additional validation of the method's effectiveness.
2. The authors should conduct hyperparameter experiments on more datasets to thoroughly analyze the impact of hyperparameters on the model's performance. This would offer a more comprehensive understanding of how different settings influence the results.

**Questions For Authors:**

Could the authors clarify whether the proposed module is applicable to all visual prompting methods, or if it is specifically suited for certain types of methods, such as token-based prompting approaches?

**Relation To Broader Scientific Literature:**

This paper introduces a plug-and-play enhancement module, TCPA, designed for existing visual prompt learning methods (such as VPT, VP, VFPT, etc.). The module works by modifying the interaction between the CLS, image tokens, and prompts in the attention mechanism, encouraging different tokens to interact with only specific prompts. This, in turn, helps the model extract more comprehensive discriminative information.

Theoretically, the paper leverages two existing theories to demonstrate that the attention matrix in current prompt learning methods is low-rank. This is further validated through experiments, which also show that the proposed method enhances the diversity of the attention matrix, encouraging different tokens to focus on a broader range of discriminative information.

**Theoretical Claims:**

The correctness of the theoretical claims in the paper is not in question. The paper relies on two established theories from existing literature to demonstrate that the self-attention matrix in current prompt learning methods is low-rank. Furthermore, through experimental analysis, it is validated that the proposed method enhances the diversity of the self-attention matrix, encouraging different tokens to focus on diverse discriminative information.

---

> ### Author Rebuttal · Authors · 2025-03-31
>
> Thank you for your appreciation of our **novelty**, **effectiveness** and **comprehensive experiments**.
>
> (The images mentioned below are available at the anonymous link: https://anonymous.4open.science/r/ICML-2025-Paper35-Rebuttal-7E9E.)
> ### Q1: More Visualizations
> 1. Thank you for your valuable feedback. We have included additional 2D and 3D attention map visualizations with more samples in the appendix.
> 2. As shown in Fig.4.1 (https://anonymous.4open.science/r/ICML-2025-Paper35-Rebuttal-7E9E/4.1.png), the same trend is observed in the additional visualizations. In existing visual prompt methods, *the attention regions of prompts are highly similar*, leading to nearly identical features being extracted from the CLS and image tokens. In contrast, our proposed TCPA module *enhances more diverse attention across prompts, CLS tokens, and image tokens*.
> 3. This is because our method selects different prompts for different tokens and performs attention-based interactions, encouraging the model to extract more diverse and comprehensive discriminative information.
> ### Q2: Hyperparameter
> 1. As shown in Fig.4.2 (https://anonymous.4open.science/r/ICML-2025-Paper35-Rebuttal-7E9E/4.2.png), we have added hyperparameter experiments *on the Dog and GTSRB datasets*, which exhibit the same trend observed on the CUB dataset.
> 2. When the prompt pool is too small, prompt diversity is limited, causing high overlap in selected prompts and making the extracted features indistinguishable. On the other hand, an excessively large prompt pool increases learnable parameters, which can lead to overfitting and decreased performance. Optimal performance is achieved with a moderate pool size.
> 3. Additionally, we have included hyperparameter experiments for the weight parameters $\lambda_i
> $ and $\lambda_c$. As shown in Fig.3.2 (https://anonymous.4open.science/r/ICML-2025-Paper35-Rebuttal-7E9E/3.2.png), the best performance is achieved when $\lambda_i=0.03$ and $\lambda_c=0.02 $. When $\lambda_i$ and $\lambda_c$ are too large, they *interfere with the learning of prompts and the classifier*, leading to performance degradation. When $\lambda_i$ and $\lambda_c$ are too small, *the indicators corresponding to the prompts cannot be effectively learned*, making it difficult to accurately match prompts to different tokens, which also results in performance degradation.
> ### Q3: Pseudocode of the Method
> 1. Thank you for your valuable suggestions.
> 2. We have added the pseudocode of the method in the appendix to clearly illustrate the proposed approach.
> ### Q4: Applicability of the Method
> 1. Existing visual prompt learning methods can be categorized into two types based on the prompt placement: image-based and token-based. *Our approach is compatible with both*.
> 2. For token-based visual prompt learning methods, our TCPA can enhance the attention interaction process, thereby improving performance.
> 3. For image-based methods, token prompts can be incorporated alongside the original approach, integrating our TCPA to provide continuous prompting during feature extraction.

---

### Official Review · Reviewer_LMp2 · 2025-03-11

**Overall Recommendation:** 3

**Summary:**

The paper introduces Token Coordinated Prompt Attention (TCPA), a novel module for visual prompting in Vision Transformers (ViTs). TCPA assigns specific prompts to CLS and image tokens, enhancing their discriminative abilities through targeted attention interactions. It uses a matching function to dynamically allocate prompts to image tokens, improving feature diversity and representation. Experiments show TCPA outperforms state-of-the-art methods, validating its effectiveness in extracting comprehensive and discriminative features.

**Claims And Evidence:**

Yes.

**Essential References Not Discussed:**

None

**Experimental Designs Or Analyses:**

The paper demonstrates the effectiveness of TCPA through comprehensive evaluations on HTA and VTAB benchmarks, consistently outperforming state-of-the-art methods like VPT and DAMVP across diverse tasks. Ablation studies clearly highlight the contributions of R-TCPA (role-level) and T-TCPA (token-level) components, with an analysis of prompt pool size. Visualizations, including t-SNE and attention maps, show that TCPA extracts more discriminative and diverse features compared to baselines. Additionally, theoretical analysis based on the low-rank properties of self-attention (Theorems 4.1 and 4.2) provides a solid foundation for TCPA's design.

**Methods And Evaluation Criteria:**

For the Token Coordinated Prompt Attention (TCPA) module, the authors propose a novel approach to disentangle prompts for CLS and image tokens, enhancing feature diversity and discriminability. However, the prompt assignment process relies on a cosine distance-based matching function, which can capture the complex relationships between tokens and prompts.

**Other Comments Or Suggestions:**

None

**Other Strengths And Weaknesses:**

Incomplete Hyperparameter Sensitivity: Only prompt pool size is explored; other hyperparameters (e.g., weighting parameters in Sec. 3.4.) are not thoroughly analyzed.

**Questions For Authors:**

How would TCPA perform if extended to the base-to-novel task [1,2] in prompt learning? It is recommended that the authors consider this direction for further research.

[1] Khattak M U, et al. Self-regulating prompts: Foundational model adaptation without forgetting[C]//ICCV. 2023: 15190-15200.
[2] Wu G, et al. Cascade prompt learning for vision-language model adaptation[C]//ECCV 2024: 304-321.

**Relation To Broader Scientific Literature:**

This paper advances the field of visual prompting by introducing the Token Coordinated Prompt Attention (TCPA) module, which enhances feature diversity and discriminability in Vision Transformers through role-specific and dynamically assigned prompts, addressing a key limitation in existing methods.

**Theoretical Claims:**

Theoretical correct.

---

> ### Author Rebuttal · Authors · 2025-03-31
>
> Thank you for your appreciation of our **novelty**, **effectiveness** and **comprehensive experiments**.
>
> (The images mentioned below are available at the anonymous link: https://anonymous.4open.science/r/ICML-2025-Paper35-Rebuttal-7E9E.)
> ### Q1: Cosine Distance-based Matching Function
> 1. We conduct a visualization experiment on the image prompts selected by different image tokens. As shown in Fig.3.1 (https://anonymous.4open.science/r/ICML-2025-Paper35-Rebuttal-7E9E/3.1.png), tokens corresponding to different parts of an object select different prompts. This indicates that our matching mechanism can recognize the different semantic information contained in tokens to some extent and assign the corresponding prompts accordingly.
> 2. During the method design process, we explored various alternative matching strategies, including Euclidean distance, Kullback-Leibler (KL) divergence, and weight prediction through a learnable MLP. As shown in the table below, among these matching strategies, cosine distance achieved the best performance.
> 3. In future research, we will further optimize the prompt matching mechanism by incorporating neighborhood token information to achieve more accurate prompt assignment.
>
> ||CUB|Dog|GTSRB|
> |-|-|-|-|
> |Euclidean Distance|89.3|91.4|93.4|
> |KL Divergence|89.2|91.3|93.6|
> |MLP|89.0|91.2|93.7
> |Cosine Distance|**89.5**|**91.5**|**94.1**|
> ### Q2: Hyperparameter
> 1. Thank you for your valuable suggestions. We conduct an ablation study on the weight hyperparameters $\lambda_i$ and $\lambda_c$ on the CUB dataset. As shown in Fig.3.2 (https://anonymous.4open.science/r/ICML-2025-Paper35-Rebuttal-7E9E/3.2.png), the best performance is achieved when $\lambda_i=0.03$ and $\lambda_c=0.02 $.
> 2. When $ \lambda_i $ and $ \lambda_c $ are too large, they *interfere with the learning of prompts and the classifier*, leading to performance degradation.
> 3. When $ \lambda_i $ and $ \lambda_c $ are too small, *the indicators corresponding to the prompts cannot be effectively learned*, making it difficult to accurately match prompts to different tokens, which also results in performance degradation.
> ### Q3: Base-to-novel Task
> 1. First, *for a fair comparison* with existing visual prompt learning methods, we conducted experiments on the HTA and VTAB benchmarks to validate the effectiveness of our approach.
> 2. Existing visual prompt learning methods typically use a ViT backbone. When adapting a pretrained ViT model to downstream tasks, a task-specific classifier must be learned *based on the number of categories in the target task*. Consequently, due to the constraints of the classifier, ViT models cannot conveniently train on base classes and generalize to novel classes as CLIP models do.
> 3. Additionally, we adapt several visual prompt learning methods, including VP, VPT, and VFPT, and our TCPA, to the CLIP model to evaluate their performance on the base-to-novel task. As shown in the table below, using the VFPT method as an example, our TCPA improves performance by **0.5%–0.9% on base classes** and by **0.5%–1.3% on novel classes**. Similarly, our TCPA also achieves consistent performance improvements on the VP and VPT methods. This improvement stems from our approach’s ability to assign different prompts to different image tokens, enabling the model to capture more comprehensive and fine-grained discriminative information for each category, thereby enhancing generalization to novel classes.
> 4. In future research, we plan to incorporate part-level modeling of learned categories to further improve the applicability and generalizability of our approach.
>
> |Methods||Caltech101|OxfordPets|Stanford_Cars|Flowers102|
> |-|-|-|-|-|-|
> |VP|Base / Novel|97.0 / 93.5|92.3 / 94.1|68.4 / 72.8|89.5 / 70.2|
> |**+TCPA**|Base / Novel|97.4 / 94.1|93.1 / 94.4|70.7 / 73.6|95.5 / 71.2|
> |VPT|Base / Novel|97.3 / 93.9|95.3 / 94.3|72.4 / 74.1|96.2 / 71.1|
> |**+TCPA**|Base / Novel|97.8 / 94.6|95.7 / **94.9**|73.7 / 74.7|96.5 / 72.3|
> |VFPT|Base / Novel|97.6 / 94.4|95.9 / 94.2|73.4 / 74.6|96.6 / 71.3|
> |**+TCPA**|Base / Novel|**98.1** / **94.9**|**96.3** / 94.8|**74.2** / **75.7**|**97.5** / **72.6**|

---

### Official Review · Reviewer_zc31 · 2025-03-13

**Overall Recommendation:** 4

**Summary:**

This paper proposes Token Coordinated Prompt Attention (TCPA) to enhance visual prompting for Vision Transformers. By disentangling and adaptively assigning prompts to different CLS and image tokens based on their distinct roles, this method effectively mitigates the limitations of conventional visual prompting and improves feature diversity and discriminability.

**Claims And Evidence:**

Yes. Figure 1 supports the claim that existing visual prompting methods usually learn and leverage the same prompt for all tokens without considering the different functionalities of CLS and image tokens, as well as the varying discriminative information conveyed by different image tokens, leading to different tokens focusing on similar regions and extracting biased discriminative information.

**Essential References Not Discussed:**

No. The related work is enough to understand the research background of this paper.

**Experimental Designs Or Analyses:**

Yes. In Section 5.5.3., there is a lack of analysis of the two weight parameters \lambda_i and \lambda_c in Equation (14). Beside, the last part of Section 5.5.4. contains redundant experimental analysis.

**Methods And Evaluation Criteria:**

Yes. The proposed method selects different prompts for different tokens and performs attention-based interactions, thereby improving the representation ability of ViT.

**Other Comments Or Suggestions:**

(1)Please revise Section 3 of the paper to ensure clarity in the methodology and consistency in notation.
(2)Please revise Section 5.5.4. of the paper to remove duplicate experimental analyses.
(3)It is recommended to provide a more detailed explanation of Equation (11) and Equation (14) and include a parameter analysis of the weight parameters in the experimental section.

**Other Strengths And Weaknesses:**

Strengths:
The proposed TCPA addresses a key limitation of existing visual prompting methods, i.e., uniform prompt interaction, improving the diversity and representational capacity of the extracted features.

Weaknesses:
(1)The method description lacks sufficient clarity in Section 3.
(2)The use of mathematical notation is somewhat inconsistent, which hinders comprehension.
(3)The optimization objective lacks a detailed explanation, and the weight parameters are not supported by a thorough parameter analysis experiment.

**Questions For Authors:**

(1)What is the purpose of the attention region corresponding to "Row 9, Column 8" in Figure 1?
(2)For the sentence “Through the equation above, we obtain the output for the image tokens (p_{j+1}^{i,d}, h_{1}^{j+1}, · · · , h_{M}^{j+1}), which, together with the previously obtained output of the CLS token c_{j+1}, serves as the input for the next MSA block B_{j+1}.” in Section 3.2, why is p_{j+1}^{i,d} not discarded.
(3)Are k_{k}^{i} and \kappa_{k}^{i} in Equation (9) representing the same notation?
(4)Does \hat{A}_{m,k} represent an element of the binarized matrix  \tilde{A}?
(5)Please explain Equation (11) and Equation (14) in detail, and add a parameter analysis of the weight parameters in the experimental section.
(6)The sentence below Equation (11) in Section 3.3, “…, we obtain the mask M^{i} corresponding to the CLS token,” seems to contain a typo. M^{i} should be written as M^{c}.
(7)There are repeated experimental analyses in Section 5.5.4.

**Relation To Broader Scientific Literature:**

Prior methods use the same prompts for all tokens without considering the distinct roles of CLS and image tokens, as well as the differences in discriminative information extracted by various image tokens. This results in the features extracted by different tokens being neither distinguishable nor comprehensive, which limits the model's performance. This paper proposes TCPA to select different prompts for different tokens and performs attention-based interactions, thereby encouraging the model to extract more diverse and comprehensive discriminative information.

**Theoretical Claims:**

In this paper, Theorem 4.1 and Theorem 4.2 do not involve specific proofs.

---

> ### Author Rebuttal · Authors · 2025-03-31
>
> Thank you for your appreciation of our **motivation**, **effectiveness** and **comprehensive experiments**.
> ### Q1: Theorem
> 1. Theorem 4.1 and Theorem 4.2 mentioned in our paper are established theories from existing work, which we have *appropriately cited*.
> 2. In their original paper, these theorems were used to analyze the rank of the attention matrix in vision models. In our paper, we reference these theorems alongside Fig.3 to illustrate that existing prompt learning methods tend to focus on overlapping information, whereas our TCPA assigns different prompts to different tokens, *enabling a more comprehensive extraction of discriminative features*.
> 3. We have now included the proof of Theorem 4.1 and Theorem 4.2 in the appendix for completeness.
> ### Q2: Hyperparameter
> Thank you for your valuable suggestions. We have included hyperparameter experiments for the weight parameters $ \lambda_i $ and $ \lambda_c $, as detailed in **Reviewer LMp2  Q2:Hyperparameter**.
> ### Q3: Section 5.5.4
> 1. In Section 5.5.4, we primarily present the t-SNE visualizations of the features extracted by both the baseline methods and our approach, further demonstrating that our method captures more comprehensive discriminative information.
> 2. We have revised and refined the last two sentences in this section to eliminate redundancy.
> ### Q4: Notations in Section 3
> 1. We have thoroughly checked the formulas in the paper. For example, we have standardized the use of *superscripts to indicate different attributes* of tokens, prompts, and indicators (i.e., whether they belong to the CLS token or image token), while *indices representing layer numbers and patch indices are consistently placed in subscripts*. For example, the notation $ \boldsymbol{h}^1_i $, which originally represented the first token in the $ i $-th layer, has been revised to $ \boldsymbol{h}_{i,1} $. Additionally, to improve clarity, we have changed $ M $, which originally represented the number of patches, to $ N $, and $ N $, which originally represented the number of network layers, to $ L $.
> 2. Furthermore, we have carefully reviewed other parts of the paper and corrected grammatical issues and typos. For example, in Eq.9, we have corrected $ \boldsymbol{k} $ to $ \boldsymbol{\kappa} $.
> ### Q5: Optimization Objective
> 1. The objective of Eq.14 is to *minimize* the distance between image tokens and their corresponding prompt indicators, ensuring that both $\sum{\mathcal{S}(\boldsymbol{h}_m, \boldsymbol{\kappa}^i_m)}$ and $\sum{\mathcal{S}(\boldsymbol{c}_j, \boldsymbol{\kappa}^c_j)}$ are as small as possible.
> 2. To enhance clarity, we have added an explanation of the optimization objective before Eq.14.
> ### Q6: Eq.11
> 1. The binarized matrix $\mathrm{\mathbf{\hat{A}}} \in \{0,1\}^{M \times N_i}$ has dimensions matching *the number of image tokens*, but in the actual model, **the CLS token needs to be considered**. In other words, the dimensions of $\mathrm{\mathbf{\hat{A}}}$ need to match *the number of image tokens plus one*. Eq.11 is designed to achieve this dimensional alignment.
> 2. To help readers better understand, we have added an explanatory note before Eq.11 in the paper.
> ### Q7: "Row 9, Column 8" in Figure 1
> 1. The first two are objects, and a background token is selected as a reference for comparison to make the experiment more comprehensive.
> 2. Existing methods focus on the same regions regardless of whether the tokens correspond to objects or the background, whereas in our approach, different object tokens attend to different object regions, and the background token focuses more on the background.
> 3. This is because our TCPA matches different prompts to tokens with different semantics for attention interactions, enabling more comprehensive extraction of discriminative information from the image.
> ### Q8: Notation $ p_{j+1}^{i,d} $ in Section 3.2
> 1. This is a typo on our part. The output $p_{j+1}^{i,d}$ for each layer should be discarded and not used for the next layer's output.
> ### Q9: Eq.9
> 1. Yes, $k_{k}^{i}$ and $\kappa_{k}^{i}$ in Eq.9 represent the same notation.
> 2. We have standardized it to $\kappa_{k}^{i}$.
> ### Q10: Notation $\hat{A}_{m,k}$
> 1. Yes, $\hat{A}_{m,k}$ represents an element of the matrix $\tilde{A}$.
> 2. We have standardized it to $\hat{A}$ and no longer use $\tilde{A}$ to represent it.
> ### Q11: Typo
> 1. We have revised $\mathrm{\mathbf{M}}^i$ to $\mathrm{\mathbf{M}}^c$ in the sentence below Eq.11 in Section 3.3.
> 2. We have also carefully checked and corrected other sections of the paper for grammatical issues and typos, such as changing "SPT" to "VPT" in Section 5.5.

---

> > ### Comment · Reviewer_zc31 · 2025-04-08
> >
> > I have reviewed the authors' rebuttal and the comments from other reviewers. I would like to maintain my positive rating.

---

> > > ### Author Response · Authors · 2025-04-08
> > >
> > > Dear reviewer zc31
> > >
> > > Thank you for your thoughtful feedback and for reconsidering our work. Your comments helped us refine the presentation and strengthen the manuscript. We truly appreciate the opportunity to clarify our approach and the time you spent reviewing our submission.
> > >
> > > Best regards,
> > >
> > > Authors

---

### Official Review · Reviewer_o6fY · 2025-03-14

**Overall Recommendation:** 3

**Summary:**

This paper introduces a token-wise prompt termed as TCPA to enrich discriminative information of tokens by assigning specific prompts to each different tokens. As a plug-and-play strategy，TCPA can be seamlessly integrate with existing prompt based methods. Experiments show that TCPA can achieve consistent performance gains across diverse benchmarks.

## update after rebuttal
The rebuttal has well addressed my concerns, so I changed my scores from 2->3. I recommend acceptance of this paper.

**Claims And Evidence:**

The claims made in the submission are supported by clear and convincing evidence.

**Essential References Not Discussed:**

None.

**Experimental Designs Or Analyses:**

Sound.

**Methods And Evaluation Criteria:**

The proposed approach contains several unresolved technical issues. The motivation is supported by previous existing works, yet the reason the proposed methods can assign different prompt to different image tokens is unclear. There is no theoretical analysis or strategy to ensure the load balance of different prompts. The eq.14 that optimizes keys of prompts looks wrong.

**Other Comments Or Suggestions:**

1.	Figure 3 is difficult to interpret due to unclear axis labels. It should be immediately discernible which axis represents keys and which represents values. Revisions to these figures may needed.
2.	The notation throughout the paper is inconsistent and poorly defined. For example, M is used to represent both the number of tokens and the final mask, which can cause confusion. The subscript c represents layers, while the subscript h represents index, which is counterintuitive.
3.	The paper contains several grammatical errors and awkward phrasings that impede clarity.
4.	Typos in 5.5 (SPT).

**Other Strengths And Weaknesses:**

See other parts.

**Questions For Authors:**

I may raise my score if all my concerns are well addressed.

**Relation To Broader Scientific Literature:**

Two existing works [1,2] served as theoretical support to this paper.
[1] Wang, S., Li, B. Z., Khabsa, M., Fang, H., and Ma, H. Linformer: Self-attention with linear complexity. arXiv preprint arXiv:2006.04768, 2020.
[2] Kim, Y., Li, Y., Moitra, A., Yin, R., and Panda, P. Do we really need a large number of visual prompts? Neural Networks, 177:106390, 2024.

**Theoretical Claims:**

No theoretical claims.

---

> ### Author Rebuttal · Authors · 2025-03-31
>
> Thank you for your appreciation of our **clearity**, **motivation** and **sound experiments**.
>
> (The images mentioned below are available at the anonymous link: https://anonymous.4open.science/r/ICML-2025-Paper35-Rebuttal-7E9E.)
> ### Q1: Load Balance of Different Prompts
> 1. Since the number of CLS and image tokens varies across different semantics, some tokens share similar meanings, while others are more distinct. As a result, prompts associated with different semantics should be selected at different frequencies. Therefore, we did not impose a strict load-balancing design for different prompts.
> 2. To ensure that every prompt is optimized rather than some prompts never being selected, we *randomly choose prompts during the first 10 epochs of model training*. In the subsequent 90 epochs, prompt selection is guided by a prompt indicator, which adjusts selection based on semantic information. This design ensures that all prompts and their corresponding indicators are optimized while maintaining differing selection frequencies for prompts associated with different semantics.
> 3. Furthermore, we visualize the selection frequency of prompts, as shown in Fig.1.1 (https://anonymous.4open.science/r/ICML-2025-Paper35-Rebuttal-7E9E/1.1.png). The results indicate that different CLS prompts are selected at frequencies **ranging from 0.13 to 0.45**, while different image prompts are selected at frequencies **ranging from 0.05 to 0.24**. This verifies that all prompts in our method are effectively utilized, with none remaining unused.
> ### Q2: Eq.14
> 1. We have carefully reviewed Eq.14 and the related Eq.9 and confirmed that our formulations are **correct**.
> 2. Eq.9, $\mathcal{S}(\boldsymbol{h}_m^{j},\boldsymbol{\kappa}^i_k)=1-\mathrm{cos}(\boldsymbol{h}_m^{j}, \boldsymbol{\kappa}^i_k)$, computes the **cosine distance** between $\boldsymbol{h}_m^{j}$ and $\boldsymbol{\kappa}^i_k$. A *higher* similarity between $\boldsymbol{h}_m^{j}$ and $\boldsymbol{\kappa}^i_k$ results in a *larger* $\mathrm{cos}(\boldsymbol{h}_m^{j},\boldsymbol{\kappa}^i_k)$, leading to a *smaller* cosine distance $\mathcal{S}(\boldsymbol{h}_m^{j},\boldsymbol{\kappa}^i_k)$.
> 3. The objective of Eq.14 is to *minimize* the distance between image tokens and their corresponding prompt indicators, ensuring that both $\sum{\mathcal{S}(\boldsymbol{h}_m,\boldsymbol{\kappa}^i_m)}$ and $\sum{\mathcal{S}(\boldsymbol{c}_j,\boldsymbol{\kappa}^c_j)}$ are as small as possible.
> 4. To enhance clarity, we have added an explanation of the optimization objective before Eq.14.
> ### Q3: Fig.3
> 1. Figure 3 in our paper visualizes the attention map in ViT, defined as $A=\text{Softmax} \left(\frac{Q K^T}{\sqrt{d_k}}\right)$.
> 2. In (b) and (d), both the x-axis and y-axis are token indices, with the y-axis corresponding to queries and the x-axis to keys. The color variations indicate the attention weights.
> 3. In (a) and (c), the z-axis is the attention weights, while both the x-axis and y-axis correspond to token indices. The difference is that the x-axis represents queries, and the y-axis represents keys. Notably, for clarity, we did not visualize all queries but instead focused on CLS tokens and a subset of prompt and image tokens.
> 4. We have revised Fig.3, explicitly annotating the meaning of each axis. The updated figure is provided in Fig.1.2 (https://anonymous.4open.science/r/ICML-2025-Paper35-Rebuttal-7E9E/1.2.png).
> ### Q4: Notation
> 1. In our original paper, $M$ represents the number of patches. It is **italic**, **uppercase**, and **not bold**, used to denote a *scalar*. In contrast, $\mathrm{\mathbf{M}}$ represents the final mask. It is **upright**, **uppercase**, and **bold**, used to denote a *tensor*.
> 2. To improve clarity, we have changed $M$, which originally represented the number of patches, to $N$, and $N$, which originally represented the number of network layers, to $L$.
> 3. The paper does not use subscripts $c$ and $h$. Instead, superscripts $c$ and $i$ are used to differentiate between CLS token-related and image token-related prompts and indicators. These are **lowercase**, **italic**, and **not bold**. Meanwhile, $\boldsymbol{h}$ represents tokens in the network, which are *vectors* and are written in **lowercase**, **italic**, and **bold**.
> 4. To enhance readability, we have standardized the use of superscripts to indicate different attributes of tokens, prompts, and indicators (belong to CLS token or image token). Meanwhile, indices indicating layer numbers and patch indices are consistently placed in subscripts. For example, the notation $\boldsymbol{h}^1_i$, which originally represented the first token in the $i$-th layer, has been revised to $\boldsymbol{h}_{i,1}$.
> ### Q5: Typo
> 1. We have corrected "SPT" to "VPT" in Sec.5.5.
> 2. Additionally, we have thoroughly checked and revised other parts of the paper for grammatical issues and typos. For example, in Eq.9, we have corrected $\boldsymbol{k} $ to $\boldsymbol{\kappa}$.

---

### Decision · Program_Chairs · 2025-05-01

**Decision:**

Accept (poster)

**Comment:**

This paper received four positive ratings, with all reviewers generally inclined to accept it. The paper introduces a Token Coordinated Prompt Attention (TCPA) module, which is concerned with the unique role of different tokens in conveying discriminative information and assigns prompts according to the different roles of different CLS and image tokens. TCPA enriches discriminative information of tokens and improves the diversity and recognizability of visual prompt features. The paper is well-organized, providing comprehensive and convincing experiments and analyses to demonstrate its effectiveness. The design of the TCPA is supported by solid theoretical foundations and experimental evidence. However, some minor errors should be corrected, such as unclear labels on the axes of the figure, inconsistent symbols, and typos. The authors have addressed the concerns raised, resolving most of the doubts. Overall, it is a good work, and the Area Chair (AC) recommends accepting the paper.